# Delineating the early transcriptional specification of the mammalian trachea and esophagus

Akela Kuwahara[1,2,3,4,5], Ace E Lewis[1,2,3,4], Coohleen Coombes[1,2,3,4,6], Fang-Shiuan Leung[1,2,3,4], Michelle Percharde[7,8], Jeffrey O Bush[1,2,3,4]*

[1]Program in Craniofacial Biology, University of California San Francisco, San Francisco, United States; [2]Department of Cell and Tissue Biology, University of California San Francisco, San Francisco, United States; [3]Institute for Human Genetics, University of California San Francisco, San Francisco, United States; [4]Eli and Edythe Broad Center of Regeneration Medicine and Stem Cell Research, University of California San Francisco, San Francisco, United States; [5]Developmental and Stem Cell Biology Graduate Program, University of California San Francisco, San Francisco, United States; [6]Department of Biology, San Francisco State University, San Francisco, United States; [7]MRC London Institute of Medical Sciences (LMS), London, United Kingdom; [8]Institute of Clinical Sciences (ICS), Faculty of Medicine, Imperial College London, London, United Kingdom

**Abstract** The genome-scale transcriptional programs that specify the mammalian trachea and esophagus are unknown. Though NKX2-1 and SOX2 are hypothesized to be co-repressive master regulators of tracheoesophageal fates, this is untested at a whole transcriptomic scale and their downstream networks remain unidentified. By combining single-cell RNA-sequencing with bulk RNA-sequencing of *Nkx2-1* mutants and NKX2-1 ChIP-sequencing in mouse embryos, we delineate the NKX2-1 transcriptional program in tracheoesophageal specification, and discover that the majority of the tracheal and esophageal transcriptome is NKX2-1 independent. To decouple the NKX2-1 transcriptional program from regulation by SOX2, we interrogate the expression of newly-identified tracheal and esophageal markers in *Sox2/Nkx2-1* compound mutants. Finally, we discover that NKX2-1 binds directly to *Shh* and *Wnt7b* and regulates their expression to control mesenchymal specification to cartilage and smooth muscle, coupling epithelial identity with mesenchymal specification. These findings create a new framework for understanding early tracheoesophageal fate specification at the genome-wide level.

*For correspondence:
jeffrey.bush@ucsf.edu

Competing interests: The authors declare that no competing interests exist.

## Introduction

Proper specification of the trachea and esophagus is critical for the function of the respiratory and digestive systems. Formation of the trachea and esophagus requires signals from the splanchnic mesenchyme which specify ventral and dorsal domains of the foregut endoderm tube, resulting in lung bud outgrowth and the initiation of physical separation of the trachea and esophagus (*Billmyre et al., 2015*; *Cardoso and Lü, 2006*; *Domyan et al., 2011*; *Goss et al., 2009*; *Harris-Johnson et al., 2009*; *Morrisey et al., 2013*; *Rankin et al., 2018*; *Rankin et al., 2016*; *Shannon et al., 1998*; *Stevens et al., 2017*). Ultimately, the tracheal epithelium is composed of multi-ciliated, secretory, and basal cells within a pseudostratified columnar monolayer which is surrounded by mesenchyme-derived ventral cartilaginous rings, and dorsal smooth muscle of the trachealis. The esophagus consists ultimately of a stratified squamous keratinized epithelium

**eLife digest** The trachea or windpipe is a tube that connects the throat to the lungs, while the esophagus connects the throat to the stomach. The trachea has cartilage rings that help to ensure clear airflow to the lungs, while the esophagus walls are lined with muscles that help to move food to the stomach. Although there are many differences between them, both the trachea and esophagus form from the same group of cells during development.

Proteins called transcription factors help to control the formation of different body parts by switching different groups of genes on and off in different subsets of cells. Existing research has suggested that a transcription factor called NKX2.1 drives trachea formation, while another, called SOX2, is important in esophagus formation. An absence of either of these two proteins is thought to be associated with serious birth defects including loss of the trachea or esophagus, or failure of the two to separate fully. How these two transcription factors interact and drive the development of the trachea and esophagus, however, is currently unclear.

Kuwahara et al. used mice to study the role of NKX2.1 and SOX2 in the formation of the trachea and esophagus. The findings identify many new genes that are active in the trachea and esophagus and reveal that NKX2.1 is not a master regulator that controls all of the genes involved in trachea formation. However, NKX2.1 does control several key genes, particularly those involved in forming cartilage in the trachea instead of muscle in the esophagus. The investigation also revealed many genes that are not controlled by NKX2.1 suggesting that other, currently unknown, systems play a major role in trachea formation.

More work is required to understand the wider genetic regulators involved in differentiating the trachea from the esophagus. The findings in this study will help researchers to understand birth defects in the trachea and esophagus that result from genetic errors. They also reveal information about gene regulation processes that are relevant to the formation of other body parts and in the context of other diseases. In the long term, they could support regenerative medicine to regrow or replace lost or damaged body parts using lab-grown stem cells.

surrounded by smooth muscle. Early failure of tracheoesophageal (TE) fate specification can result in an unseparated common foregut endoderm tube, producing common congenital pathologies known as tracheoesophageal fistula (TEF) or tracheal agenesis (TA) (*Billmyre et al., 2015*; *Morrisey and Hogan, 2010*; *Sher and Liu, 2016*). Other anomalies of foregut development, such as tracheomalacia and congenital tracheal stenosis, involve the improper formation of the adjacent tracheal mesenchyme-derived cartilage and smooth muscle (*Morrisey and Hogan, 2010*; *Sher and Liu, 2016*). While foregut malformations in humans are common, we know very little of the genetic causes of these defects, in part due to a paucity of information of normal tracheoesophageal transcriptional patterning.

The earliest marker of tracheal and lung fate is the transcription factor NKX2-1 (TTF1) which is expressed in the ventral foregut in a pattern complementary to the dorsally enriched expression of the transcription factor SOX2 (*Guazzi et al., 1990*; *Minoo et al., 1999*; *Mizuno et al., 1991*; *Que et al., 2007*). Disruption of *Nkx2-1* in mice resulted in upregulation of SOX2 in the ventral endoderm and differentiation of the adjacent mesenchyme into smooth muscle rather than tracheal cartilage (*Minoo et al., 1999*; *Que et al., 2007*). Conversely, hypomorphic disruption of *Sox2* in mice resulted in upregulation of dorsal NKX2-1 and a conversion of the stratified esophageal epithelium to a simple columnar epithelium surrounded by smooth muscle that histologically resembles that of the trachea (*Que et al., 2007*; *Teramoto et al., 2019*). Similarly, knockdown of SOX2 in human induced pluripotent stem cell (hiPSC)-derived dorsal foregut cells resulted in upregulation of NKX2-1, and forced expression of SOX2 in hiPSC-derived ventral foregut cells repressed NKX2-1 (*Trisno et al., 2018*). Together these data have given rise to a model in which NKX2-1 and SOX2 form a co-repressive master regulatory switch to define tracheal and esophageal cell fates (*Billmyre et al., 2015*; *Domyan et al., 2011*; *Que et al., 2007*; *Teramoto et al., 2019*; *Trisno et al., 2018*). The regulatory programs downstream of NKX2-1 and SOX2 are not known and, therefore, the extent to which each promotes or represses tracheal and esophageal cell fates is not clear. Moreover, beyond these two transcription factors, we currently know very little about the

transcriptional identity of the early dorsoventral endodermal populations that ultimately give rise to the trachea and esophagus.

The mechanisms coupling epithelial and mesenchymal fate specification in the trachea and esophagus are not well understood, but involve epithelial to mesenchymal signaling. For example, loss of WNT signaling from the endoderm to the tracheal mesenchyme results in a loss of tracheal cartilage and a corresponding expansion of smooth muscle (*Hou et al., 2019*; *Kishimoto et al., 2019*; *Snowball et al., 2015*). SHH signaling regulates smooth muscle specification in multiple contexts (*Huycke et al., 2019*; *Mao et al., 2010*) and loss of SHH signaling from the airway and intestinal epithelium results in loss of smooth muscle formation (*Kim et al., 2015*; *Litingtung et al., 1998*; *Pepicelli et al., 1998*) and mispatterning of tracheal cartilage (*Miller et al., 2004*; *Sala et al., 2011*). Thus, while WNT and SHH signaling are critical for foregut mesenchymal differentiation, how these signals are transcriptionally regulated in the tracheal and esophageal epithelium is currently unknown.

In this study, we dissect the transcriptional regulation of tracheal and esophageal fate specification by combining multiple genomic approaches. By single cell RNA-sequencing (scRNA-seq) we define the transcriptional identity of the trachea, esophagus, and lung at their initial stages of development, and identify new and robust markers of tracheoesophageal specification. We then dissect the NKX2-1 regulatory program that specifies TE identity using our scRNA-seq datasets, in combination with bulk RNA-sequencing of $Nkx2-1^{-/-}$ mutant tracheas, and NKX2-1 chromatin immunoprecipitation and sequencing (ChIP-seq) of wild type tracheas. We discover a previously unknown NKX2-1-independent transcriptional program that encompasses the majority of the newly-identified tracheal and esophageal transcriptomes. We assay the NKX2-1 transcriptional program in functional compound mouse mutant experiments to test whether NKX2-1 regulates these TE genes through repression of SOX2 or independently of SOX2. These data uncover a role for NKX2-1 in regulating epithelial-to-mesenchymal signaling, thereby coupling TE epithelial identity with cartilage and smooth muscle fate specification. This study therefore establishes a new framework for understanding key regulators of early cell fate specification in the trachea and esophagus.

## Results

### Single cell transcriptomics identifies dorsoventral populations of the foregut

To understand how the trachea and esophagus are specified on a transcriptome-wide scale, we performed droplet-based single-cell RNA sequencing (scRNA-seq) on E10.5 mid-separation and E11.5 post-separation dissected mouse foregut epithelial cells. We generated 6407 single-cell transcriptomes at E10.5, comprising 5 cell clusters, and 10,493 single-cell transcriptomes at E11.5, comprising 7 cell clusters, and visualized these clusters using Uniform Manifold Approximation and Projection (UMAP) dimensional reduction (*Becht et al., 2019*; *Stuart et al., 2018*; *Figure 1a,b*). We delineated the dorsoventral axis in our scRNA-seq data at E10.5 and E11.5 by projecting the expression levels of *Nkx2-1* and *Sox2* on the UMAP (*Figure 1—figure supplement 1c,d*). Similarly, *Sox9*, a marker of developing lung epithelium (*Herriges et al., 2012*; *Perl et al., 2005*; *Rockich et al., 2013*), marked a subset of *Nkx2-1*-positive respiratory cells, enabling us to distinguish distal lung from trachea (*Figure 1—figure supplement 1c,d*). Using differential expression analysis for each cell cluster and RNAscope fluorescent in-situ hybridization, we defined cell clusters as lung, trachea, esophagus, and pharynx (*Figure 1a–l''*, *Figure 1—figure supplement 2a–m*, *Figure 1—figure supplement 3a–c*, *Figure 1—figure supplement 4a*) as well as a cluster corresponding to the ultimobranchial body, a derivative of the pharyngeal endoderm that gives rise to the follicular cells of the thyroid (*Nilsson and Fagman, 2017*; *Figure 1—figure supplement 4b*). Within the E11.5 lung, we identified a cluster with unique expression of known distal lung markers such as *Bmp4* (*Weaver et al., 1999*; *Figure 1—figure supplement 3c*, *Figure 1—source data 1*), demonstrating that the proximodistal lung axis can be identified by markers in our dataset. Signatures of proliferation also divided both the trachea and lung clusters at E11.5 (*Figure 1b*, *Figure 1—source data 1*).

Our scRNA-seq data revealed a wealth of genes previously unknown to mark cell types of the ventral and dorsal foregut prior to TE separation (*Figure 1c*, *Figure 1—source data 1*), as well as new genes that distinguish the trachea, lung, and esophagus after TE separation (*Figure 1—figure*

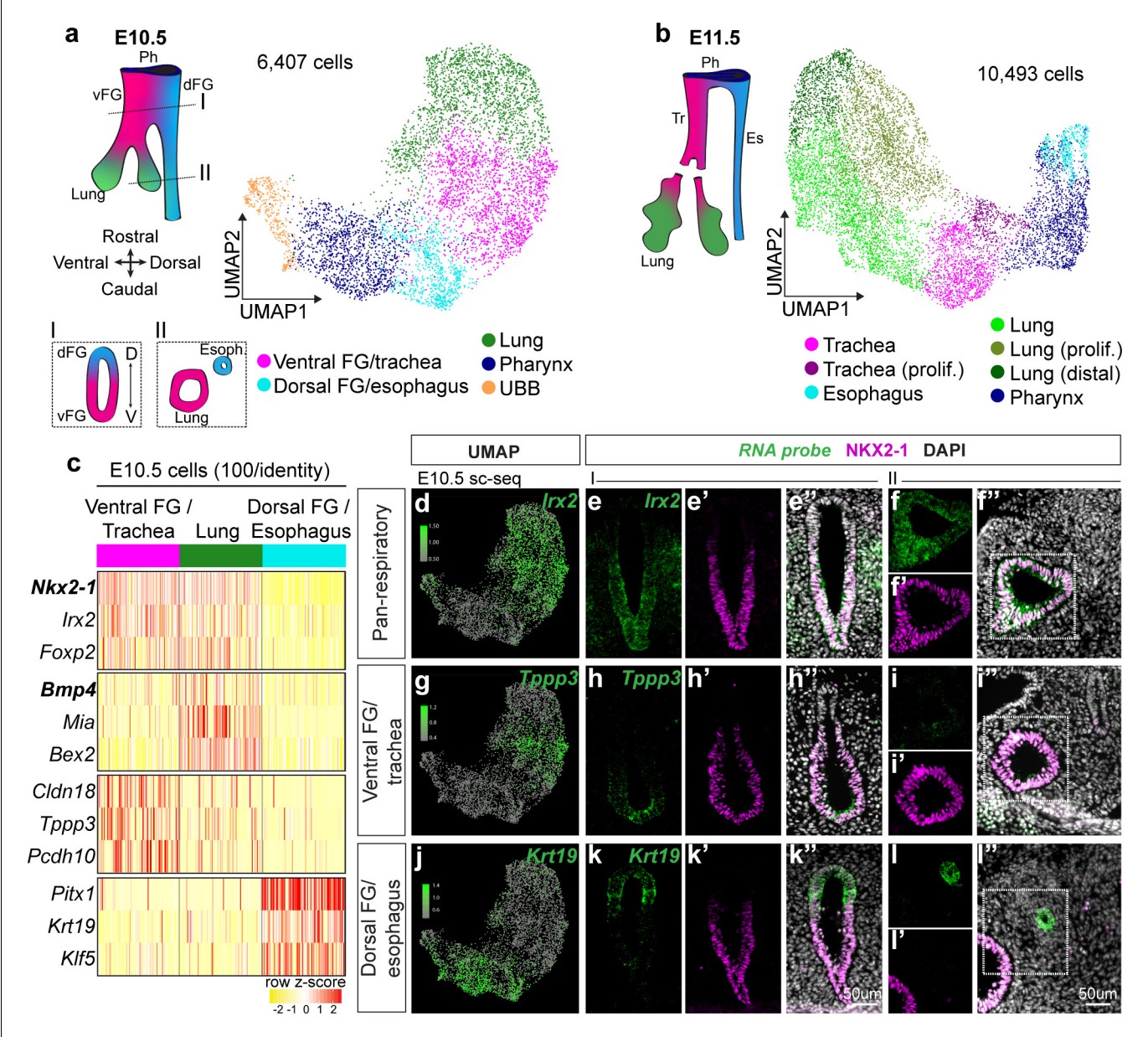

**Figure 1.** Single-cell transcriptomics of the developing foregut epithelium identifies distinct dorsoventral populations. Dissected, FACS-purified foregut epithelial cells were subjected to droplet-based single-cell RNA sequencing at E10.5 and E11.5. (a) UMAP representation of 6,407 cells identified at E10.5 and (b) 10,493 cells identified at E11.5. Colors represent cell populations identified using shared nearest neighbor clustering. vFG: ventral foregut, dFG: dorsal foregut, Ph: pharynx, UBB: ultimobranchial body, Tr: trachea, Es: esophagus. (c) Heatmap of selected marker gene expression across 100 E10.5 cells each of ventral foregut/trachea, lung, and esophagus/dorsal foregut as identified by scRNA-seq. Selected genes are markers of respiratory, lung, ventral FG/trachea, and dorsal FG/esophagus cells (top to bottom). n = 1 biological replicate/timepoint with 20 pooled embryos at E10.5 and 28 pooled embryos at E11.5. See also: *Figure 1—source data 1*. (d-l) RNA localization of pan-respiratory marker gene *Irx2*, ventral FG/trachea marker gene *Tppp3*, and dorsal FG/esophagus marker gene *Krt19* identified from differential expression analysis of cell populations in scRNA-seq data at E10.5. First column shows projection of RNA expression level as determined by scRNA-seq on UMAP. Second-fourth columns show RNA expression of marker gene (green) and NKX2-1 expression (magenta) in the undivided region of E10.5 embryonic foreguts (I) as indicated by the positions of the schematic in a). Last two columns show staining in the lung and distal esophagus (II) as indicated by the positions of the schematic in a). All images were captured at 20X magnification and displayed at the same scale. Scale bar = 50 um.

The online version of this article includes the following source data and figure supplement(s) for figure 1:

**Source data 1.** Top 20 markers of foregut cell populations identified by scRNA-seq of E10.5 and E11.5 foreguts.

**Figure supplement 1.** Quality control metrics and characterization of single-cell RNA sequencing experiments.

**Figure supplement 2.** Additional cell type-specific markers of dorsoventral populations identified by scRNA-seq at E10.5.

*Figure 1 continued on next page*

*Figure 1 continued*

**Figure supplement 3.** Cell type-specific markers of trachea, esophagus, and lung populations identified by scRNA-seq at E11.5.
**Figure supplement 4.** Newly identified esophageal marker genes are more dorsally restricted than SOX2.
**Figure supplement 5.** Validation of pharynx and UBB cluster identities.

*supplement 3c*, *Figure 1—source data 1*). To visualize their spatial expression, we performed RNA-scope for several genes marking each of these cell types in the undivided E10.5 foregut and lung (*Figure 1d–l*; *Figure 1—figure supplement 2a–m*), and the E11.5 trachea, esophagus, and lung (*Figure 1—figure supplement 3*). In all cases, RNAscope analysis confirmed our scRNA-seq finding and provided additional information about the spatial patterns of marker gene expression. We found many genes, including *Klf5*, *Krt19*, *Dcn*, *Pitx1*, and *Lrig1* that exhibited expression specifically within the dorsal foregut/esophageal cells (*Figure 1c,j–l''*; *Figure 1—figure supplement 2k–m''*, *Figure 1—figure supplement 3*, *Figure 3—figure supplement 1b*). Interestingly, within the foregut endoderm, all of these genes were more dorsally restricted than SOX2 and were more precisely complementary to NKX2-1, indicating that they may serve as better markers of the esophagus during its early development (*Figure 1—figure supplement 4*). Notably, we also discovered genes such as *Tppp3*, *Pcdh10*, *Ly6h*, and *Cldn18* that exhibited enrichment in the ventral foregut at E10.5 (*Figure 1c,g–i''*, *Figure 1—figure supplement 2e-j'*) and specifically marked the trachea and proximal airway at E11.5 (*Figure 1—figure supplement 3*). Our discovery of this gene class indicates that the trachea and proximal airway are actively transcriptionally specified and at least somewhat distinct from the lung during early respiratory development. All tracheal and esophageal markers we identified and validated were dorsoventrally restricted in the common foregut of E10.5 embryos prior to physical separation of the trachea and esophagus, consistent with extensive fate specification before TE separation (*Billmyre et al., 2015*). Together, these data uncover a multitude of previously unknown genes that define initial dorsoventral patterning of the foregut and the earliest stages of the trachea, lung, and esophagus.

## Identification of the NKX2-1 transcriptional program and an independent program of TE specification

Given our discovery of a broad network of previously unknown genes expressed during early TE specification, we sought to examine how tracheoesophageal fates are dysregulated upon *Nkx2-1* loss at the transcriptome-wide scale. We performed RNA-sequencing (RNA-seq) of dissected and FACS-purified foregut epithelium from E11.5 *Nkx2-1*[-/-] and wild-type (WT) embryos. Differential expression analysis between *Nkx2-1*[-/-] and WT foregut epithelium identified 257 NKX2-1-dependent genes, with 109 genes upregulated and 148 genes downregulated in *Nkx2-1*[-/-] foreguts (*Figure 2a*, *Figure 2—source data 1*). We examined the expression of these NKX2-1-dependent genes in our scRNA-seq dataset at E11.5 to determine, on a global-scale, where they were expressed in the developing foregut. We found that genes that were upregulated in *Nkx2-1*[-/-] mutants were enriched in cells of the esophagus and pharynx (*Figure 2b*, *Figure 2—figure supplement 1a*), and genes that were downregulated in *Nkx2-1*[-/-] mutants were enriched in tracheal and lung cells (*Figure 2c*, *Figure 2—figure supplement 1b*). Together, these data define the NKX2-1 transcriptional program in early TE development and support the hypothesis that, within this program, NKX2-1 positively regulates tracheal genes and negatively regulates esophageal genes.

Surprisingly, our scRNA-seq dataset identified many genes that mark tracheal and esophageal cells that did not appear to exhibit a change in expression in our *Nkx2-1*[-/-] mutant RNA-seq analysis (*Figure 1—source data 1*, *Figure 2—source data 1*). We examined the spatial expression of several of these NKX2-1-independent genes using an *Nkx2.5-cre* strain which mediates recombination in the ventral foregut (*Figure 3—figure supplement 1a*; *Stanley et al., 2004*) to generate *Nkx2-1*[lox/lox]; *Nkx2.5*[cre/+] (Nkx2-1-TrKO) embryos lacking NKX2-1 in the trachea. All Nkx2-1-TrKO embryos we examined exhibited a complete failure of foregut separation (n = 8/8 embryos). Using RNAscope, we found *Irx2*, *Ly6h*, and *Nrp2* to be tracheal-specific and maintained in the ventral epithelium of the unseparated foregut tube in E11.5 Nkx2-1-TrKO embryos (*Figure 3a–f'*; *Figure 3—figure supplement 1b*). Likewise, we found *Dcn*, *Ackr3*, and *Meis2* to be esophageal-specific and maintained in the dorsal region of the common foregut tube in Nkx2-1-TrKO foreguts (*Figure 3g–l'*; *Figure 3—*

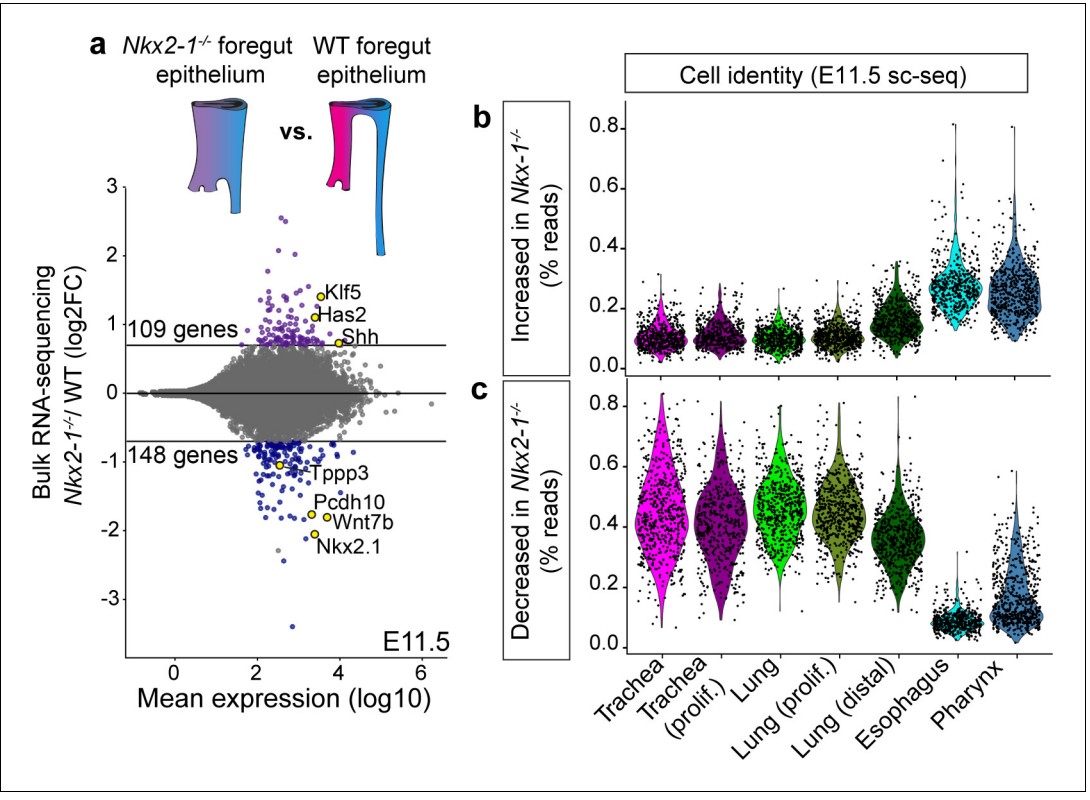

**Figure 2.** Transcriptomic analysis of *Nkx2-1*−/− foreguts reveals NKX2-1 regulated tracheal and esophageal transcriptional programs. (a) Dissected, FACS-purified epithelium from E11.5 *Nkx2-1*−/− and WT foreguts was sequenced and analyzed for differential gene expression. 109 genes were increased and 148 genes were decreased in *Nkx2-1*−/− foreguts compared to WT (DESeq2, log2FC > 0.7, padj < 0.05). Labeled genes show examples of tracheal genes that decrease in *Nkx2-1*−/− mutants (*Tppp3, Pcdh10, Wnt7b, Nkx2-1*), and esophageal genes that increase in *Nkx2-1*−/− mutants (*Klf5, Has2, Shh*). n = 3 biological replicates with two pooled embryos/ replicate. See also: ***Figure 2—source data 1***. (b-c) Combined expression of all genes that increase (b) or decrease (c) in *Nkx2-1*−/− foreguts as a percentage of all reads in E11.5 scRNA-seq data. Cells are grouped by their assigned cluster and clusters were subsampled to 600 cells/cluster for visualization.

The online version of this article includes the following source data and figure supplement(s) for figure 2:

**Source data 1.** Differentially expressed genes between E11.5 *Nkx2-1*−/− and WT foregut epithelium identified with bulk RNA-seq.

**Figure supplement 1.** NKX2-1-dependent gene expression in E11.5 scRNA-seq.

*figure supplement 1b*). Immunofluorescent staining also showed that the esophageal genes LRIG1 and PITX1 were maintained in the dorsal region of *Nkx2-1*−/− foreguts (***Figure 3—figure supplement 1b***). These data indicate the presence of an NKX2-1-independent transcriptional program within the developing trachea and esophagus, and suggest that *Nkx2-1*−/− mutant foreguts do not undergo a complete tracheal-to-esophageal fate conversion.

We therefore examined the extent to which tracheoesophageal patterning is independent of NKX2-1 by generating bulk transcriptional profiles of WT tracheal and esophageal epithelium at E11.5 to compare with our *Nkx2-1*−/− mutant bulk RNA-seq dataset (***Figure 3m***). Differential expression analysis identified 1126 genes enriched in the trachea and 809 genes enriched in the esophagus of WT embryos (***Figure 3n***, ***Figure 3—source data 1***). Notably, only 11% of tracheal-enriched genes and 6% of esophageal-enriched genes in WT foreguts were affected by loss of NKX2-1, and the majority of tracheal- or esophageal-enriched genes were unchanged in *Nkx2-1*−/− foreguts (***Figure 3n***). Together, these findings define relevant NKX2-1 transcriptional targets during early tracheoesophageal development and uncover a significant NKX2-1-independent gene regulatory program.

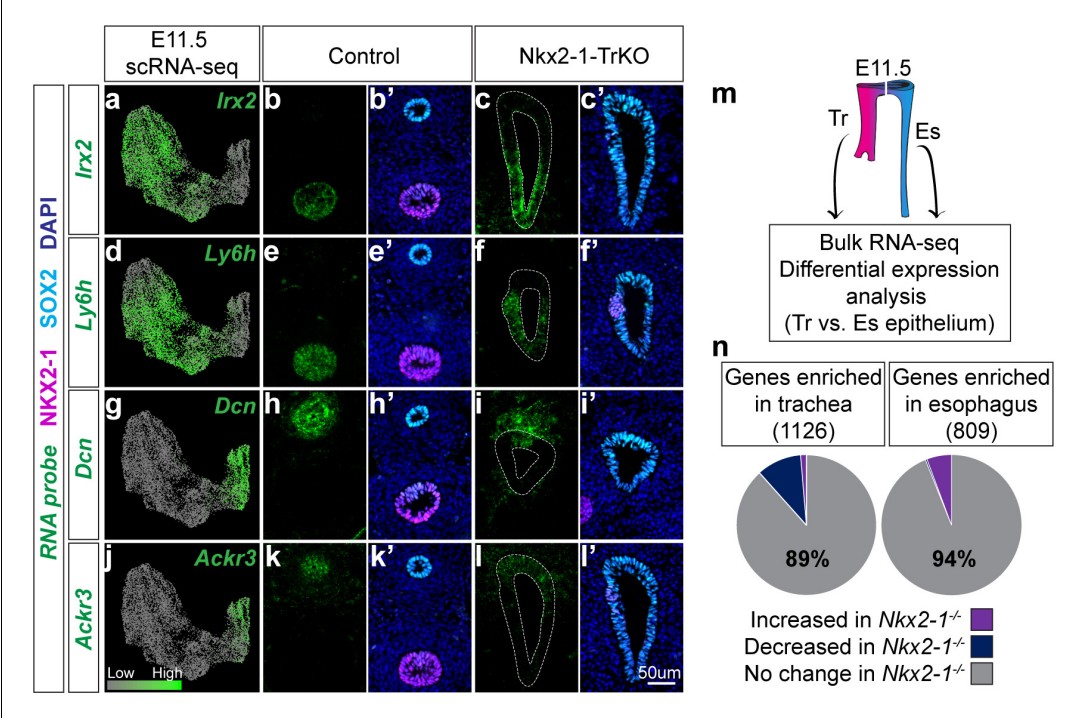

**Figure 3.** Identification of an NKX2-1-independent tracheal and esophageal program. (a-l) RNA localization of NKX2-1-independent tracheal (*Irx2, Ly6h*) and esophageal (*Dcn, Ackr3*) makers identified by scRNA-seq and *Nkx2-1⁻/⁻* mutant RNA-seq data at E11.5. First column shows projection of RNA expression level as determined by scRNA-seq on UMAP. Second-fifth columns show RNA localization (green) in E11.5 control and Nkx2-1-TrKO embryos with immunofluorescent staining of NKX2-1 (magenta) and SOX2 (cyan). 8/8 Nkx2-1-TrKO embryos had TEF phenotype, n = 3 embryos/ staining combination. All images were captured at 20X magnification and displayed at the same scale. Scale bar = 50 um. (m) Schematic of experimental procedures for RNA-seq and differential expression analysis of WT trachea and esophagus for comparison with *Nkx2-1⁻/⁻* mutant RNA-seq data. n = 3 biological replicates with four pooled embryos/replicate. See also: *Figure 3—source data 1*. (n). NKX2-1-dependent genes as a portion of genes enriched in WT trachea (left) and esophagus (right). NKX2-1-independent genes make up 89% of tracheal-enriched genes and 94% of esophageal enriched genes.

The online version of this article includes the following source data and figure supplement(s) for figure 3:

**Source data 1.** Differentially expressed genes between E11.5 WT trachea and esophagus epithelium identified with bulk RNA-seq.

**Figure supplement 1.** *Nkx2.5-cre* recombination, additional validation of NKX2-1-independent genes.

## NKX2-1 directly regulates foregut genes, with variable dependency on SOX2

To identify direct targets of NKX2-1 during TE specification, we performed NKX2-1 chromatin immunoprecipitation followed by sequencing (ChIP-seq) on dissected E11.5 WT trachea. We identified 15,861 genomic regions (peaks) shared between two biological replicates that showed NKX2-1 binding (*Figure 4a*, FDR < 0.00001) and were centrally enriched for the known NKX2-1 motif (*Figure 4b*, p=3.4e-37). We next looked closely at NKX2-1 binding at or near the loci of select NKX2-1-dependent tracheal and esophageal genes. We observed NKX2-1 binding at the *Nkx2-1* promoter and five binding sites within 10 kb of the *Nkx2-1* gene, consistent with previous data suggesting that NKX2-1 autoregulates its own expression (*Nakazato et al., 1997*; *Oguchi and Kimura, 1998*; *Tagne et al., 2012*), and also observed binding of NKX2-1 at the *Sox2* promoter and four binding sites within the *Sox2* locus, suggesting direct repression of *Sox2* by NKX2-1 (*Figure 4c*). In addition, NKX2-1 binding was observed at the promoter of genes such as *Pcdh10*, *Tppp3*, and *Klf5* that we identified as specific markers of the tracheal and esophageal lineages by scRNA-seq (*Figure 4c*), suggesting that these genes may also be direct targets of NKX2-1 regulation. Genome-wide comparison of NKX2-1-bound genes with the *Nkx2-1⁻/⁻* de-regulated transcriptome revealed that NKX2-1-dependent genes were associated with NKX2-1 ChIP-seq peaks at a higher frequency than observed at random (*Figure 4d*, Fisher's exact test, p<0.0001). Furthermore, when we divided NKX2-1-dependent genes

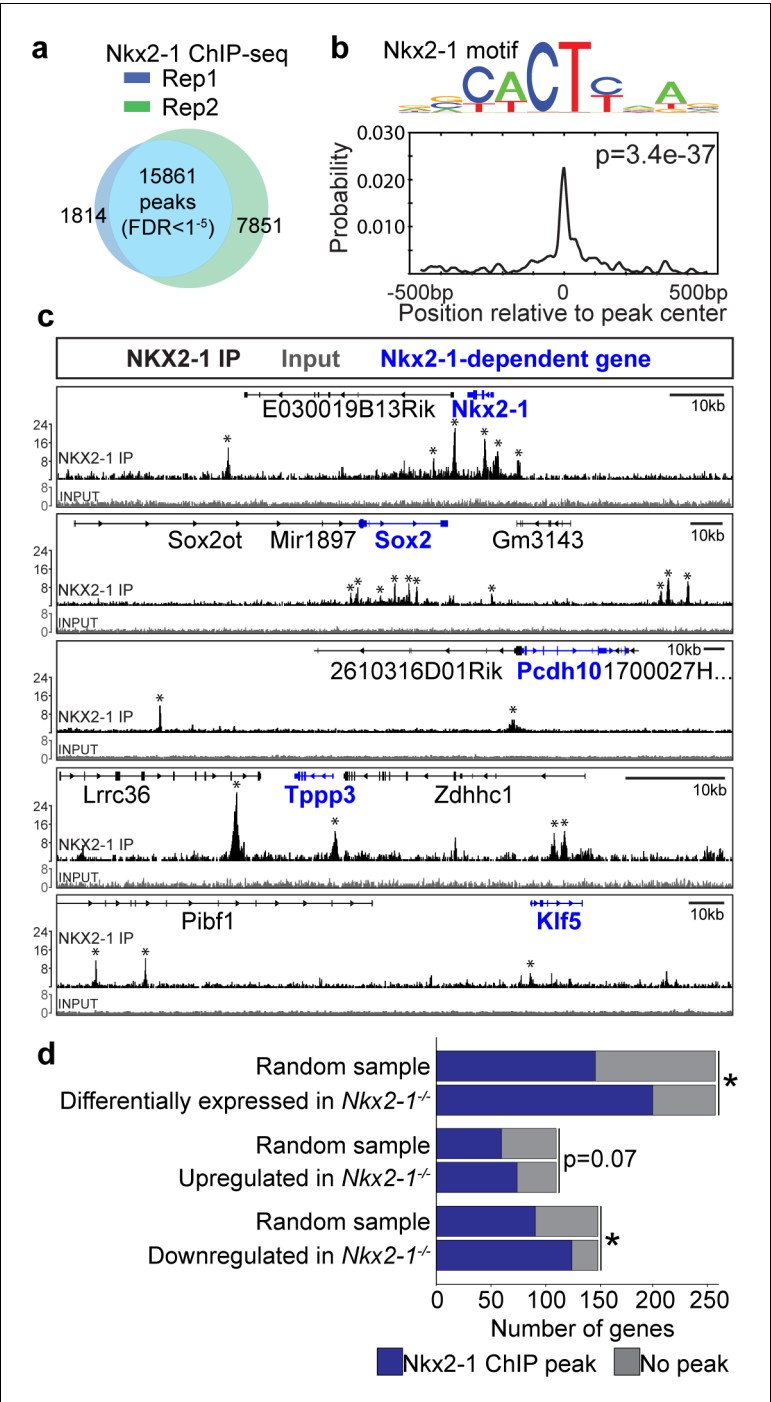

**Figure 4.** NKX2-1 binds directly to tracheoesophageal genes in the developing foregut.  (a) ChIP-seq for NKX2-1 in E11.5 foreguts identified 15,861 NKX2-1-bound genomic regions (peaks) shared between replicates (FDR < 0.00001). n = 2 biological replicates with 175 pooled embryos/replicate. (b) Motif analysis of NKX2-1 ChIP-seq data shows peaks are enriched for the NKX2-1 motif (p=3.4e-37). (c) NKX2-1 ChIP-seq (black) and input (grey) tracks near loci of select NKX2-1-dependent genes (blue) *Nkx2-1, Sox2, Pcdh10, Tppp3,* and *Klf5* visualized in IGV. Input and NKX2-1 IP tracks are displayed at the same linear scale, as indicated by IGV Data Range on the y-axis. Horizontal scale bar = 10 kb. (d) Genome-wide comparison of NKX2-1 ChIP-seq with *Nkx2-1$^{-/-}$* mutant RNA-seq. Overlap of NKX2-1 ChIP-seq associated genes with NKX2-1-dependent genes (top), genes increased in *Nkx2-1$^{-/-}$* mutants (middle) and genes decreased in *Nkx2-1$^{-/-}$* mutants (lower). *asterisk = p < 0.0001, Fisher's exact test.

into those that were upregulated or downregulated in *Nkx2-1*[-/-] mutants, we found that genes that are downregulated in *Nkx2-1*[-/-] mutants are more frequently associated with NKX2-1 ChIP-seq peaks (*Figure 4d*, Fisher's exact test, p<0.0001). These data suggest that whereas NKX2-1 is a direct positive regulator of tracheal-specific genes, repression of esophageal-specific genes may more often be indirect.

Based on previous studies, NKX2-1 indirect regulation may be mediated through its repression of *Sox2* (*Billmyre et al., 2015*). Indeed, given the well-established genetically co-repressive relationship of SOX2 and NKX2-1 in the foregut (*Domyan et al., 2011*; *Que et al., 2007*; *Teramoto et al., 2019*; *Trisno et al., 2018*), it is challenging to determine whether the transcriptional changes we observed in *Nkx2-1*[-/-] mutants are solely due to the loss of NKX2-1 or also due to the subsequent gain of SOX2. Thus, we devised a genetic strategy to uncouple NKX2-1 and SOX2 regulation of tracheal and esophageal genes by generating *Nkx2-1; Sox2* compound mutant embryos. To achieve this, we again utilized *Nkx2.5-cre* to generate *Nkx2-1^{lox/lox}; Nkx2.5^{cre/+}* (Nkx2-1-TrKO) embryos lacking NKX2-1, and *Nkx2-1^{lox/lox}; Sox2^{lox/lox}; Nkx2.5^{cre/+}* (Nkx2-1-TrKO; Sox2-TrKO) embryos lacking both NKX2-1 and SOX2 in the ventral foregut/trachea cells (*Figure 5a–c*). Similar to the Nkx2-1-

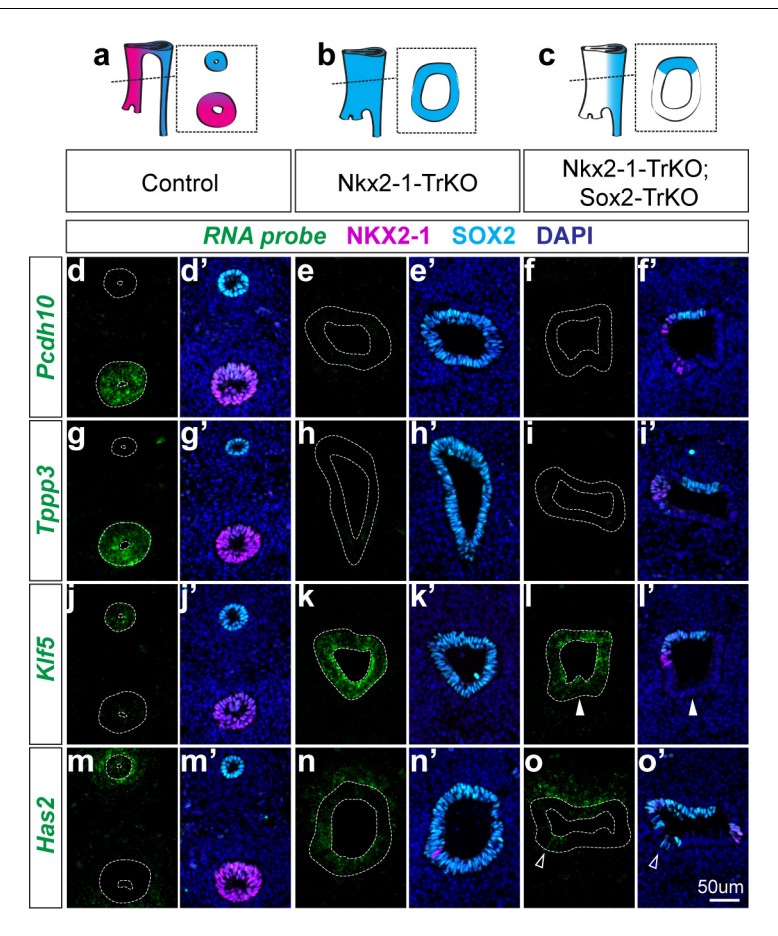

**Figure 5.** NKX2-1 regulates target genes in a SOX2-dependent and independent manner. (a-c) Schematic of *Nkx2-1; Sox2* compound mutant analysis phenotypes and resulting NKX2-1 and SOX2 expression patterns in E11.5 control embryos (a), Nkx2-1-TrKO mutants (b), Nkx2-1-TrKO; Sox2-TrKO mutants (c). 8/8 Nkx2-1-TrKO embryos and 6/6 Nkx2-1-TrKO; Sox2-TrKO embryos examined had TEF phenotype. (d-f) RNA localization of NKX2-1-dependent, SOX2-independent gene *Pcdh10* in control (d), Nkx2-1-TrKO (e), and Nkx2-1-TrKO; Sox2-TrKO (f) embryos with immunofluorescent staining of NKX2-1 (magenta) and SOX2 (cyan). (g–i) Tracheal, NKX2-1-dependent, SOX2-independent gene *Tppp3*. (j–l) Esophageal, NKX2-1-dependent, SOX2-independent gene *Klf5*. Solid arrowheads indicate SOX2-negative, *Klf5*-positive ventral cell. (m–o) Esophageal, NKX2-1-dependent, SOX2-dependent gene *Has2*. Arrowheads indicate ventral SOX2-positive, *Has2*-positive cells. n = 3 embryos/staining combination. All images were captured at 20X magnification and displayed at the same scale. Scale = 50 um.

TrKO embryos, all Nkx2-1-TrKO; Sox2-TrKO embryos we examined exhibited a complete failure of foregut separation (n = 6/6 embryos). We then examined NKX2-1-regulated genes that were determined by our scRNA-seq analysis to be markers of E11.5 trachea or esophagus. The expression of the tracheal-specific genes *Pcdh10* and *Tppp3* was decreased in Nkx2-1-TrKO and Nkx2-1-TrKO; Sox2-TrKO ventral foreguts indicating that their expression requires NKX2-1 but not upregulation of SOX2 (*Figure 5d–i'*), Conversely, expression of the esophageal-specific genes *Klf5* and *Has2* was upregulated in the ventral foreguts of Nkx2-1-TrKO, as predicted from our RNA-seq analysis (*Figure 5j–o'*). Interestingly, whereas *Klf5* expression was also increased in the ventral foregut of Nkx2-1$^{tr/tr}$; Sox2$^{tr/tr}$ mutants (arrowhead *Figure 5l,l'*), *Has2* expression was not, with the exception of a few cells that retain SOX2 expression ventrally (arrowhead in *Figure 5o,o'*). Thus, while *Klf5* and *Has2* were both repressed by NKX2-1, upregulation of *Has2* in the ventral foregut of Nkx2-1-TrKO embryos appears to also depend on the upregulation of SOX2 in this region. Together, these results revealed that NKX2-1 regulation results from both direct activation of tracheal genes and repression of esophageal genes, as well as the indirect suppression of target genes through the repression of *Sox2*, illustrating the complex relationship between these two transcription factors with opposing expression patterns.

## NKX2-1 regulates *Wnt7b* and *Shh* to couple tracheal endoderm identity to mesenchymal cell fate

The apparent fate transformation of *Nkx2-1$^{-/-}$* mutant trachea to esophagus is supported by the conversion of ventral mesenchymal cell fates from tracheal cartilage to smooth muscle (*Minoo et al., 1999*; *Que et al., 2007*; *Yuan et al., 2000*). We confirmed previous reports of a dramatic reduction and disorganization of tracheal cartilage with expansion of the smooth muscle in *Nkx2-1$^{-/-}$* mutants, and additionally observed malformation of the thyroid and cricoid cartilage (n = 3/3 embryos, *Figure 6a,o,p*). Our finding that the endoderm in *Nkx2-1$^{-/-}$* mutants did not exhibit a complete fate transformation led us to ask whether we could uncover the specific NKX2-1 targets that impact epithelial to mesenchymal signaling amongst this more limited set of candidates. Notably, *Wnt7b* expression decreased and *Shh* increased in *Nkx2-1$^{-/-}$* foregut epithelium compared to WT (*Figure 2a*), and both signaling genes have known functions in tracheal cartilage and smooth muscle formation. In WT E11.5 and E13.5 embryos, *Wnt7b* is expressed in the tracheal epithelium and *Shh* is expressed more strongly in the esophageal epithelium (*Gerhardt et al., 2018*; *Litingtung et al., 1998*; *Rajagopal et al., 2008*; *Snowball et al., 2015*; *Figure 6c,f,i,m*). We examined our NKX2-1 ChIP-seq data to determine whether NKX2-1 binds directly to *Wnt7b* and *Shh* in the trachea and observed binding at three sites 20 kb and 60 kb upstream and 20 kb downstream of the *Wnt7b* gene. Though a previous study detected binding of NKX2-1 to the *Wnt7b* promoter in a lung epithelial cell line (*Weidenfeld et al., 2002*), we instead observe binding at three sites 20 kb and 60 kb upstream and 20 kb downstream of the *Wnt7b* gene. We also observed two NKX2-1 ChIP-seq peaks within the *Shh* gene, and two additional peaks 50 kb and 150 kb upstream of *Shh*, consistent with the possibility of direct regulation by NKX2-1 (*Figure 6b*). Together, these data indicate that *Wnt7b* and *Shh* are targets of NKX2-1 regulation in the developing foregut.

To determine whether NKX2-1 regulates *Wnt7b* and *Shh* through a change in SOX2 expression, we examined *Nkx2-1; Sox2* compound mutants. *Wnt7b* expression was lost in both Nkx2-1-TrKO and Nkx2-1-TrKO; Sox2-TrKO foreguts, suggesting that NKX2-1 regulates *Wnt7b* independently of regulation by SOX2 (*Figure 6c–e'*). However, the increase in *Shh* expression in the ventral foregut observed in Nkx2-1-TrKO embryos was not observed in Nkx2-1-TrKO; Sox2-TrKO foreguts (*Figure 6f–h'*), indicating that SOX2 is required for upregulation of *Shh* in the ventral foregut. Thus, *Wnt7b* expression is positively regulated by NKX2-1 but not subject to regulation by SOX2, whereas the dorsally restricted expression of *Shh* is mediated by the combined action of NKX2-1 and SOX2.

To determine whether NKX2-1 regulation of *Wnt7b* and *Shh* impacts epithelial-mesenchymal signaling, we first examined the expression of *Wnt7b* and downstream canonical WNT signaling transcriptional target *Axin2* in E11.5 and E13.5 *Nkx2-1$^{-/-}$* and WT foreguts. Whereas in WT embryos *Axin2* expression was localized to the ventral tracheal mesenchyme where SOX9+ cartilage progenitor cells are found, in *Nkx2-1$^{-/-}$* mutants *Axin2* expression was decreased and correlated with disorganized SOX9 expression (*Figure 6i–l*, *Figure 6—figure supplement 1a*). This decrease in *Axin2* expression was more dramatic in the caudal foregut, consistent with the increased severity of cartilage specification defects in this region seen in skeletal preparations (*Figure 6a*) and as visualized by

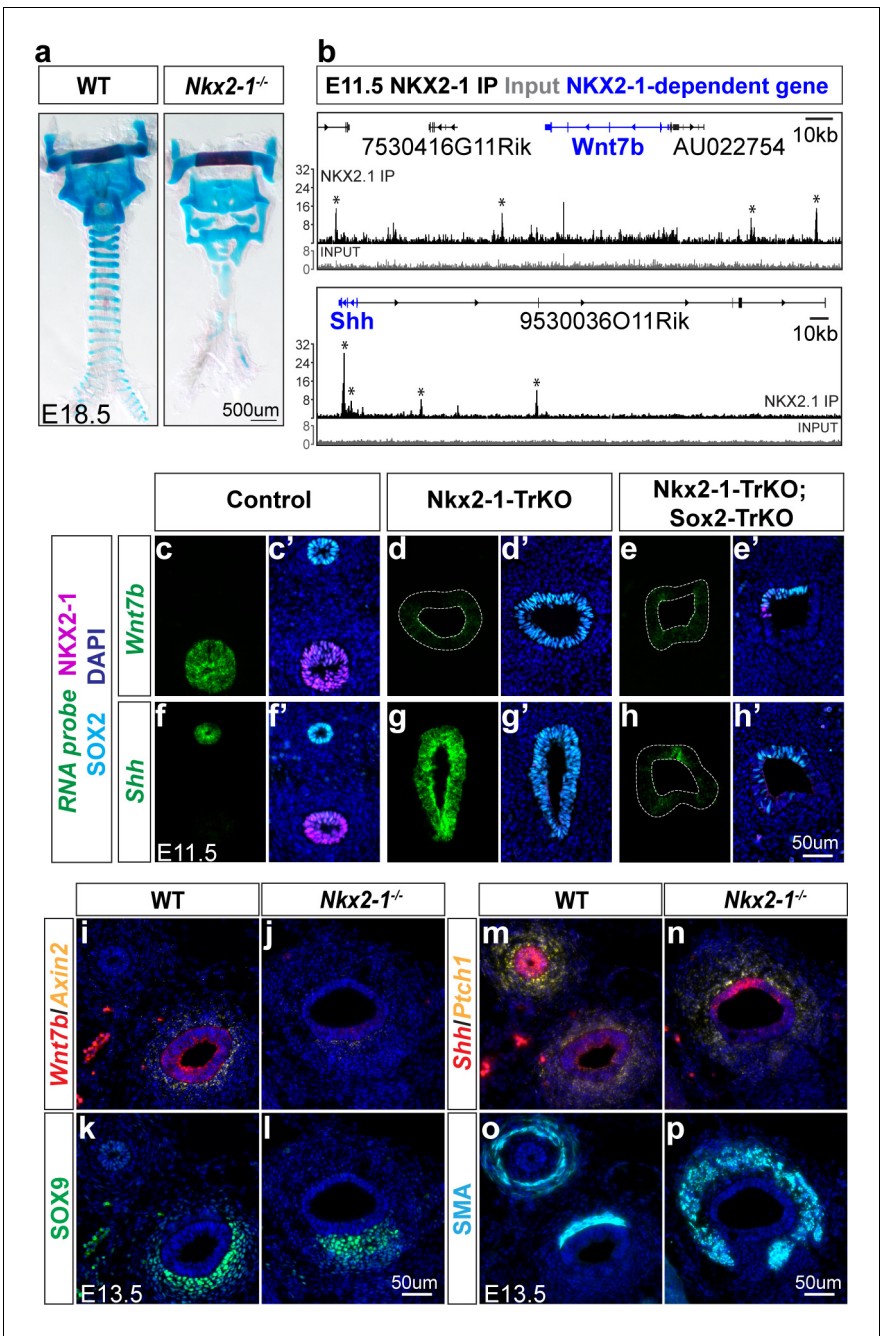

**Figure 6.** NKX2-1 regulates mesenchymal specification via *Wnt7b* and *Shh*-mediated epithelial-mesenchymal crosstalk. (a) Alcian blue staining of tracheal cartilage in E18.5 control and *Nkx2-1⁻/⁻* embryos. 3/3 embryos examined exhibited similar cartilage phenotype. (b) ChIP-seq of direct NKX2-1 binding near *Wnt7b* and *Shh* loci. Input and NKX2-1 IP tracks are displayed at the same linear scale, as indicated by IGV Data Range on the y-axis. Horizontal scale bar = 10 kb. (c–h) RNA localization (green) of *Wnt7b* (c–e) and *Shh* (f–h) in control (left), Nkx2-1-TrKO (middle), and Nkx2-1-TrKO; Sox2-TrKO (right) E11.5 embryos with immunofluorescent staining of NKX2-1 (magenta) and SOX2 (cyan). (i–j) RNA localization of *Wnt7b* (red) and *Axin2* (yellow) in WT and *Nkx2-1⁻/⁻* E13.5 foreguts. 3/3 embryos display phenotype. (k–l) SOX9 staining (green) of cartilage progenitors in WT and *Nkx2-1⁻/⁻* E13.5 foreguts. 3/3 embryos display phenotype. (m–n) RNA localization of *Shh* (red) and *Ptch1* (yellow) in WT and *Nkx2-1⁻/⁻* E13.5 foreguts. 2/2 embryos display phenotype. (o–p) Smooth muscle actin (SMA) staining (cyan) of smooth muscle in WT and *Nkx2-1⁻/⁻* E13.5 foreguts. 3/3 embryos display phenotype. All images were captured at 20X magnification and images within each staining panel are at the same scale. Scale = 50 um.

The online version of this article includes the following figure supplement(s) for figure 6:

*Figure 6 continued on next page*

Figure 6 continued

**Figure supplement 1.** NKX2-1 regulation of epithelial-mesenchymal signaling.

SOX9 localization in *Nkx2-1*<sup>-/-</sup> foreguts (*Figure 6—figure supplement 1c–e*). We next examined the expression of *Shh* and downstream transcriptional target *Ptch1* in *Nkx2-1*<sup>-/-</sup> mutant and WT foreguts. In E11.5 and E13.5 *Nkx2-1*<sup>-/-</sup> foreguts, increased expression of *Shh* in the ventral foregut epithelium compared to WT was mirrored by an increase in *Ptch1* expression in the surrounding mesenchyme (*Figure 6m,n*; *Figure 6—figure supplement 1b*). This increase in *Ptch1* expression also corresponded with the increase in smooth muscle as visualized by ACTA2 (i.e. α-SMA) surrounding *Nkx2-1*<sup>-/-</sup> foreguts compared to WT (*Figure 6m–p*, S8c,f,g). Thus, our data show that in addition to regulating a subset of genes that define tracheoesophageal epithelial identity at this stage, NKX2-1 and SOX2 regulate epithelial-mesenchymal crosstalk via WNT and SHH signaling to instruct mesenchymal differentiation, explaining this aspect of the observed TE fate transformation in *Nkx2-1*<sup>-/-</sup> mutant foreguts (*Figure 7*).

## Discussion

NKX2-1 is a critical regulator of respiratory development across vertebrates (*Minoo et al., 1999*; *Rankin et al., 2018*; *Small et al., 2000*; *Yuan et al., 2000*), but how it directs TE specification was unknown. It has been previously demonstrated in adenocarcinoma and in the adult distal lung, that loss of NKX2-1 leads to adoption of a gastric transcriptional phenotype (*Little et al., 2019*; *Snyder et al., 2013*). Further, mesenchymal cues that induce lung and trachea development converge to establish ventral NKX2-1 expression, and *Nkx2-1*<sup>-/-</sup> mutant embryos exhibit dramatic epithelial and mesenchymal phenotypes. Together, these findings have led to the proposal that NKX2-1 is a master regulator of initial tracheal fate specification (*Billmyre et al., 2015*; *Domyan et al., 2011*; *Goss et al., 2009*; *Harris-Johnson et al., 2009*; *Morrisey and Hogan, 2010*; *Que et al., 2007*). By performing scRNA-seq profiling on the endoderm at the onset of tracheal and esophageal development, and comparing this profile to *Nkx2-1*<sup>-/-</sup> mutant embryos, we find that NKX2-1 loss

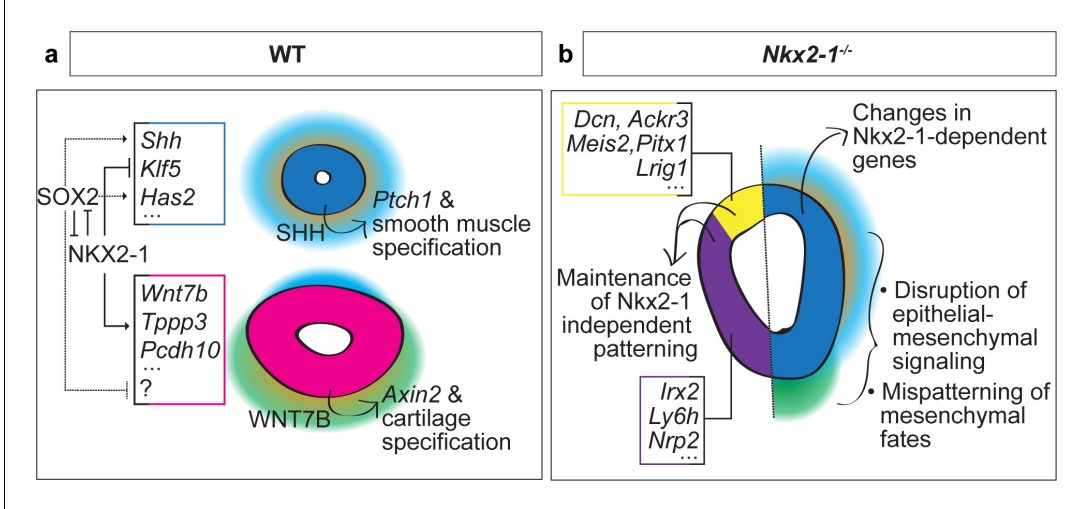

**Figure 7.** Proposed model of NKX2-1-dependent and -independent tracheoesophageal specification. (a) Schematic of NKX2-1 and SOX2 regulation of gene expression in WT trachea and esophagus, and epithelial-mesenchymal signaling downstream of NKX2-1. NKX2-1 negatively regulates esophageal genes *Shh*, *Klf5* and *Has2*, and positively regulates tracheal genes *Wnt7b*, *Tppp3*, and *Pcdh10*. SOX2 is required for expression of *Shh* and *Has2*. NKX2-1 regulation of *Shh* and *Wnt7b* influences mesenchymal SHH and WNT response required for smooth muscle and tracheal cartilage development. (b) Schematic of *Nkx2-1*<sup>-/-</sup> mutant phenotype. Maintenance of the NKX2-1-independent transcriptional program includes dorsal expression of *Dcn*, *Ackr3*, *Meis2*, *Pitx1*, and *Lrig1* and ventral expression of *Irx2*, *Ly6h*, and *Nrp2*. NKX2-1-dependent transcriptional changes include increased expression of esophageal genes and decreased expression of tracheal genes in the ventral foregut, accompanied by changes in epithelial-mesenchymal signaling and mesenchymal differentiation.

does not cause a transcriptome-wide tracheal-to-esophageal conversion, but rather results in changes in expression of a key subset of dorsoventrally restricted genes that can explain apparent fate-conversion phenotypes in the mesenchyme (*Figure 7*).

The functional significance of the NKX2-1-independent tracheal program, and how it is established, remains to be explored. It is possible that mesenchymal signals such as WNT and BMP induce other tracheal transcriptional regulators in parallel. Interestingly, the ISL1 transcription factor has been recently identified to exhibit ventrally-restricted expression in the trachea and is required for normal NKX2-1 expression (*Kim et al., 2019*). Further, our data reveal that *Isl1* is independent of NKX2-1 (*Figure 2—source data 1*), indicating that it resides hierarchically upstream of NKX2-1. It remains to be determined, however, the extent to which ISL1 regulates tracheal transcriptional identity. Our profiling of tracheoesophageal specification, combined with genomic analyses of additional mouse mutants, provide an exciting opportunity for further exploration of additional regulators of tracheoesophageal specification within the foregut epithelium.

Our scRNA-seq profiling reveals many new markers of dorsoventral foregut identity beyond *Nkx2-1* and *Sox2*. This of particular importance for human iPSC/ESC-derived models where *Nkx2-1* and *Sox2* expression are often used to indicate respiratory or esophageal/gut fate decisions (*Dye et al., 2015*; *Hawkins et al., 2017*; *Trisno et al., 2018*; *Zhang et al., 2018*). Our data suggest that in addition to using *Nkx2-1* as a marker of respiratory fate, differentiation protocols might be more thoroughly characterized using a panel of NKX2-1-dependent and -independent respiratory markers. Further, though we have not explored it in depth, we also provide here the first scRNA-seq dataset characterizing the lung at early stages of branching morphogenesis, which may additionally aid in more precise characterization of early lung development. Our study therefore provides a rich resource for expanding our understanding of early cell fate decisions in stem cell models of human foregut and lung development.

By defining the NKX2-1-dependent transcriptomic profile, we elucidate how NKX2-1 regulates initial tracheal and esophageal development at the genome-wide level. A dual repressive genetic relationship with SOX2 has been previously demonstrated, and our ChIP-seq data support direct repression of *Sox2* by NKX2-1. We find, however, that direct binding of NKX2–1 occurs more often at genes that are normally expressed in the trachea, and loss of tracheal gene expression in *Nkx2-1$^{-/-}$* mutants is not indirectly mediated by upregulation of SOX2. Together, these findings broadly support a model in which NKX2-1 more often positively regulates the tracheal gene expression program directly. While we demonstrate that SOX2 is a mediator of NKX2-1-indirect regulation of some genes, SOX2 targets in the foregut have not been identified at the transciptomic level, and it is therefore unclear the extent to which SOX2 controls TE fates. Additionally, our identification of other NKX2-1-dependent transcription factors such as KLF5 provides candidates potentially mediating NKX2-1 indirect repression of esophageal genes.

Tracheal cartilage and smooth muscle phenotypes in *Nkx2-1$^{-/-}$* embryos support a TE fate conversion phenotype (*Minoo et al., 1999*; *Que et al., 2007*); here we provide a likely explanation of these phenotypes and reveal how NKX2-1 may regulate development of the tracheal cartilage and restrict smooth muscle formation. Normal tracheal cartilage specification requires WNT signaling from the tracheal epithelium to the mesenchyme (*Hou et al., 2019*; *Kishimoto et al., 2019*; *Snowball et al., 2015*). We find that NKX2-1 positively regulates *Wnt7b* resulting in activation of a canonical WNT-signaling response in the adjacent ventral tracheal mesenchyme. A recent report suggested that epithelial-mesenchymal WNT signaling during cartilage specification occurs through NKX2-1-independent mechanisms (*Kishimoto et al., 2019*). Our results differ from this finding but do support the possibility that an NKX2-1 independent mechanism also contributes to tracheal cartilage specification, as some disorganized cartilage still forms in *Nkx2-1$^{-/-}$* mutants.

In addition to regulating WNT signaling, we found that NKX2-1 loss resulted in upregulation of *Shh* and increased SHH signal reception in the ventral mesenchyme. As HH signaling is a key regulator of smooth muscle formation in multiple contexts, and its disruption also results in perturbation of tracheal cartilage formation (*Huycke et al., 2019*; *Mao et al., 2010*; *Miller et al., 2004*; *Sala et al., 2011*), it is likely that this upregulation also contributes to disruption of tracheal cartilage and expansion of smooth muscle formation. Notably, similar to the other trachea and esophagus markers we examined, NKX2-1 regulation of *Wnt7b* does not depend on SOX2, whereas the upregulation of *Shh* in *Nkx2-1$^{-/-}$* mutants does. This fits with the broader model of NKX2-1 directly activating a

defined tracheal program, but more often repressing an esophageal program through ventral repression of SOX2.

While few genetic causes of foregut anomalies in humans have been identified, human stem cell models have indicated that key players in respiratory and gut specification are conserved between mice and humans (*Ostrin et al., 2018*; *Trisno et al., 2018*). Therefore, future work to understand genetic causes of foregut malformations in humans including tracheoesophageal fistula (TEF), tracheal agenesis, esophageal atresia, tracheomalacia, and tracheal stenosis will benefit greatly from the improved understanding of normal tracheal and esophageal specification presented in this study. Together, these findings dramatically advance our understanding of the earliest stages of tracheoesophageal development, while revising our perspective on the role of NKX2-1 in this process.

# Materials and methods

**Key resources table**

| Reagent type (species) or resource | Designation | Source or reference | Identifiers | Additional information |
|---|---|---|---|---|
| Strain, strain background (mouse) | CD1 | Harlan/Envigo | Cat#: 030 | |
| Genetic reagent (mouse) | Nkx2-1$^{fl/fl}$ | *Kusakabe et al., 2006* | MGI: 3653645 | |
| Genetic reagent (mouse) | Sox2$^{fl/fl}$ | *Shaham et al., 2009* | MGI: 4366453 | |
| Genetic reagent (mouse) | Tmem163$^{Tg}$ (Actin$^{cre}$) | *Lewandoski et al., 1997* | MGI: 2176050 | |
| Genetic reagent (mouse) | Nkx2.5cre | *Stanley et al., 2004* | MGI: 2448972 | |
| Antibody | α-NKX2-1 (rabbit polyclonal) | Millipore | Cat#: 07601 | 1:200 for IF, 5ug for ChIP |
| Antibody | α-SOX2 (goat polyclonal) | Neuromics | Cat#: GT15098 | 1:250 for IF |
| Antibody | α-SOX9 (Rabbit polyclonal) | Santa Cruz | Cat#: sc-20095 | 1:250 for IF |
| Antibody | α-PITX1 (Rabbit polyclonal) | NovusBio | Cat: NBP188644 | 1:200 for IF |
| Antibody | α-LRIG1 (Goat polyclonal) | R&D Systems | Cat#: AF3688 | 1:250 for IF |
| Antibody | α-SMA (Rabbit polyclonal) | Abcam | AB5694 | 1:300 for IF |
| Antibody | α-SMA-cy3 (mouse monoclonal) | Sigma | C6198 | 1:300 for IF |
| Antibody | α-EpCAM (rat polyclonal) | BioLegend | Cat#: 118219 | 1:200 for FACS |
| Sequence-based reagent | RNAscope probe *mm-Irx2* | Advanced Cell Diagnostics | Cat#: 519901 | |
| Sequence-based reagent | RNAscope probe *mm-Wnt7b* | Advanced Cell Diagnostics | Cat#: 401131 | |
| Sequence-based reagent | RNAscope probe *mm-Pcdh10* | Advanced Cell Diagnostics | Cat#: 477781-C3 | |
| Sequence-based reagent | RNAscope probe *mm-Tppp3* | Advanced Cell Diagnostics | Cat#: 586631 | |
| Sequence-based reagent | RNAscope probe *mm-Ly6h* | Advanced Cell Diagnostics | Cat#: 587811 | |
| Sequence-based reagent | RNAscope probe *mm-Krt19* | Advanced Cell Diagnostics | Cat#: 402941 | |

*Continued on next page*

Continued

| Reagent type (species) or resource | Designation | Source or reference | Identifiers | Additional information |
|---|---|---|---|---|
| Sequence-based reagent | RNAscope probe *mm-Klf5* | Advanced Cell Diagnostics | Cat#: 444081 | |
| Sequence-based reagent | RNAscope probe *mm-Foxe1* | Advanced Cell Diagnostics | Cat#: 509641 | |
| Sequence-based reagent | RNAscope probe *mm-Crabp1* | Advanced Cell Diagnostics | Cat#: 474711-C3 | |
| Sequence-based reagent | RNAscope probe *mm-Bmp4* | Advanced Cell Diagnostics | Cat#: 401301-C2 | |
| Sequence-based reagent | RNAscope probe *mm-Nrp2* | Advanced Cell Diagnostics | Cat#: 500661 | |
| Sequence-based reagent | RNAscope probe *mm-Dcn* | Advanced Cell Diagnostics | Cat#: 413281-C3 | |
| Sequence-based reagent | RNAscope probe *mm-Has2* | Advanced Cell Diagnostics | Cat#: 465171-C2 | |
| Sequence-based reagent | RNAscope probe *mm-Meis2* | Advanced Cell Diagnostics | Cat#: 436371-C3 | |
| Sequence-based reagent | RNAscope probe *mm-Ackr3* | Advanced Cell Diagnostics | Cat#: 482561-C2 | |
| Sequence-based reagent | RNAscope probe *mm-Axin2* | Advanced Cell Diagnostics | Cat#: 400331-C3 | |
| Sequence-based reagent | RNAscope probe *mm-Shh* | Advanced Cell Diagnostics | Cat#: 314361 | |
| Sequence-based reagent | RNAscope probe *mm-Ptch1* | Advanced Cell Diagnostics | Cat#: 402811-C2 | |
| Sequence-based reagent | Nkx2-1-geno1 | This paper | PCR primer | 5'-CTA-GGG-AGG-CTA-GGA-ACT-CGG-3' |
| Sequence-based reagent | Nkx2-1-geno2 | This paper | PCR primer | 5'-CCG-ACC-CAC-GTA-GAG-CC-3' |
| Sequence-based reagent | Nkx2-1-geno3 | This paper | PCR primer | 5'-CTC-TTA-TCT-GGG-ATC-GCC-TGA-G-3' |
| Sequence-based reagent | Sox2-flox-geno1 | *Steevens et al., 2019* | PCR primer | 5'-TGG-AAT-CAG-GCT-GCC-GAG-AAT-CC-3' |
| Sequence-based reagent | Sox2-flox-geno2 | *Steevens et al., 2019* | PCR primer | 5'-TCG-TTG-TGG-CAA-CAA-GTG-CTA-AAG-C-3' |
| Sequence-based reagent | Sox2-flox-geno3 | *Steevens et al., 2019* | PCR primer | 5'-CTG-CCA-TAG-CCA-CTC-GAG-AAG-3' |
| Sequence-based reagent | Cre-geno1 | *Liang et al., 2005*; *Vauclair et al., 2005* | PCR primer | 5'-GTT-CGC-AAG-AAC-CTG-ATG-GAC-A-3' |
| Sequence-based reagent | Cre-geno2 | *Liang et al., 2005*; *Vauclair et al., 2005* | PCR primer | 5'-CTA-GAG-CCT-GTT-TTG-CAC-GTT-C-3' |
| Commercial assay or kit | MicroChIP Diapure columns | Diagenode | Cat#: C03040001 | DNA extraction (ChIP) |
| Commercial assay or kit | Microplex Library Preparation Kit v2 | Diagenode | Cat#: C05010012 | ChIP library prep |
| Commercial assay or kit | Rneasy Micro kit | Qiagen | Cat# 74004 | RNA extraction (bulk) |
| Commercial assay or kit | SMARTer Stranded Total RNAseq kit V2 | Takara | Cat#: 634411 | Bulk RNAseq library prep |
| Commercial assay or kit | RNA 6000 Pico Bioanalyzer kit | Agilent | Cat# 5067-1513 | Bioanalyzer (RNA) |
| Commercial assay or kit | High Sensitivity DNA Bioanalyzer kit | Agilent | Cat#: 5067-4626 | Bioanalyzer (DNA) |

*Continued on next page*

*Continued*

| Reagent type (species) or resource | Designation | Source or reference | Identifiers | Additional information |
|---|---|---|---|---|
| Commercial assay or kit | QuBit dsDNA HS Assay kit | Invitrogen | Cat#: Q32854 | DNA quantification |
| Commercial assay or kit | Chromium Single Cell ' Reagent Kit V2 | 10X Genomics | Cat#: PN-120237, PN-120236, PN-120262 | Single cell RNA-seq |
| Other | Sytox Blue nuclei acid stain | Thermo | Cat: S11348 | Live/dead stain for FACS, use at 1uM |
| Other | TSAplus Fluoresceine | Akoya Biosciences | NEL741001KT | RNAscope, 1:1500 |
| Other | TSAplus Cy3 | Akoya Biosciences | NEL744001KT | RNAscope, 1:1500 |
| Other | TSAplus Cy5 | Akoya Biosciences | NEL745001KT | RNAscope, 1:1500 |
| Other | DynaBeads, protein G | Invitrogen | Cat#: 10003D | ChIP |
| Other | AMPure XP beads | Beckman Coulter | Cat#: A63881 | Library cleanup |
| Other | TrypLE Express | Thermo | Cat# 12604013 | Tissue dissociation |

## Mice

All animal procedures were performed at the University of California San Francisco (UCSF) under approval from the UCSF Institutional Animal Care and Use Committee (mouse protocol # AN164190). Mouse embryos were collected from pregnant females via cesarean section at the described timepoint following observation of a vaginal plug. Noon the day of the plug was considered embryonic day 0.5. For single cell sequencing experiments, timed-pregnant CD1 female mice were obtained from Harlan/Envigo (Cat: 030) and embryos were staged using somite counts. For mutant analysis, the following alleles were used: *Nkx2-1$^{lox/lox}$* (MGI: 3653645), *Sox2$^{lox/lox}$* (MGI: 4366453), *Nkx2.5-Cre* (MGI: 2448972), Actin-Cre (MGI: 2176050).

## Immunofluorescence

Mouse embryos were dissected at E10.5 or E11.5 in cold PBS and fixed overnight in 4% paraformaldehyde at 4C. For cryopreservation, embryos were subjected to a sucrose gradient of 12.5% sucrose in PBS for 8 hr, followed by 25% sucrose in PBS overnight at 4°C, and embedded in OCT. Tissue sections of 12 um were cut using a cryostat and used for immunofluorescence, or in-situ hybridization followed by immunofluorescence with standard protocols. Primary antibodies used for immunofluorescence were: NKX2-1 (Millipore 07601, 1:200), SOX2 (Neuromics GT15098, 1:250), SOX9 (Santa Cruz, sc-20095, 1:250), LRIG1 (R&D, AF3688, 1:200), PITX1 (NovusBio, NBP188644, 1:250).

## In-situ hybridization

For in-situ hybridization, 12 μm cryosections were stained using the RNAscope Multiplex Fluorescent Reagent Kit v2 (Advanced Cell Diagnostics, cat# 323100) with the following adjustments to the manufacturer's protocol: antigen retrieval step was bypassed, protease step used ProteasePlus for 10 min. Following in-situ hybridization, slides were washed 2x in PBS and subjected to immunofluorescent staining as described in 'Immunofluorescence' above. Probes used for in-situ hybridization against mouse RNA were obtained from Advanced Cell Diagnostics as follows: *Irx2* (519901), *Wnt7b* (401131), *Pcdh10* (477781-C3), *Tppp3* (586631), *Ly6h* (587811), *Krt19* (402941), *Klf5* (444081), *Foxe1* (509641), *Crabp1* (474711-C3), *Bmp4* (401301-C2), *Nrp2* (500661), *Dcn* (413281-C3), *Has2* (465171-C2), *Meis2* (436371-C3), *Ackr3* (482561-C2), *Axin2* (400331-C3), *Shh* (314361), *Ptch1* (402811-C2).

## Skeletal preparation

Skeletal preps were performed as previously described (*Martin et al., 1995*).

## Dissociation and FACS of embryonic tissue

Foregut tissue was dissected in cold PBS and dissociated to single cells using TrypLE Express (phenol-red free, Thermo cat# 12604013) at 37°C for 5 min, followed by trituration for 1–3 min at 37°C. Cells were washed twice with FACS buffer (2 mM EDTA and 5% fetal bovine serum in phenol-red

free HBSS). To identify epithelial cells, cells were stained with PerCP/Cy5.5 anti-mouse CD326/EpCAM (BioLegend, cat# 118219, used at 1:100) at 4°C for 30 min followed by two washes with FACS buffer. Cells were resuspended in FACS buffer with Sytox Blue nucleic acid stain (Thermo, S11348, used at 1 µM) to stain dead cells, and passed through a 35 µm cell strainer. Cells were sorted using a BD FACS Aria II. Single live epithelial cells were collected after size selection and gating for Sytox-negative, EpCAM-positive cells. For scRNA-seq, cells were sorted into EDTA-free FACS buffer and processed as described below. For bulk-RNA-seq, cells were sorted directly into RNA lysis buffer (Qiagen RNeasy Micro kit, cat# 74004) with 1% beta-mercaptoethanol and processed as described below.

## Single-cell RNA sequencing

Foregut tissue was dissected from 20 embryos at E10.5 (6-9ts) and 28 embryos at E11.5 (16-20ts). To ensure for representation of tracheal and esophageal cells at E11.5, lung tissue was separated from E11.5 foreguts and processed in parallel. Tissue from each timepoint was pooled and single-cell suspension and epithelial purification was performed as described above. 25,000 live epithelial cells from each sample were loaded into individual wells for single-cell capture using the Chromium Single Cell 3' Reagent Kit V2 (10X Genomics). Library preparation for each sample was also performed using the Chromium Single Cell 3' Reagent Kit V2, and each sample was given a unique i7 index. Libraries were pooled and subjected to sequencing in a single lane of an Illumina Nova-Seq6000. Sequencing data were processed, and downstream analysis performed as described below.

## RNA sequencing

For bulk RNA-sequencing experiments, whole foregut tissue was dissected from E11.5 *Nkx2-1*$^{-/-}$ or WT embryos, and lungs were removed at the time of dissection. For RNA-sequencing of wild type (WT) trachea and esophagus, trachea and esophagus were manually separated at the time of dissection. Foreguts of individual embryos were dissociated, stained, and sorted as described above. Each biological replicate consisted of RNA pooled from two *Nkx2-1*$^{-/-}$ or WT embryos, with a total of 3 biological replicates from different litters. RNA was purified using the RNeasy Micro kit (Qiagen, cat# 74004) and quantification was performed using the RNA 6000 Pico kit (Agilent, cat# 5067–1513) on an Agilent 2100 Bioanalyzer. RNA-sequencing libraries were prepared from 4 ng of input RNA using the SMARTer Stranded Total RNAseq kit V2 (Takara, cat# 634411) with 13 amplification cycles. Library size and quality was checked using an Agilent 2100 Bioanalyzer with the High Sensitivity DNA kit (Agilent, cat# 5067–4626), and library concentration was determined with the QuBit dsDNA HS Assay kit (Invitrogen, cat# Q32854). Libraries were normalized to 7 nM, pooled, and sequenced across two lanes of an Illumina HiSeq 4000 to generate 50 base pair single-end reads. Data processing and downstream analysis was performed as described below.

## Chromatin immunoprecipitation and sequencing

For ChIP-seq experiments, tracheas were dissected from WT E11.5 mouse embryos and cross-linked with 1% cold PFA for 9 min. Crosslinking was stopped with glycine for a final concentration of 0.125M. The crosslinked tissue was washed 2x in PBS and stored at −80°C. For each replicate 175 trachea were pooled and the tissue was thawed and dissociated in cold PBS by passing through a 25G needle until fully dissociated. The cells were lysed in 500 µl lysis buffer (50 mM Tris-HCl pH8, 2 mM EDTA pH8, 0.1% NP-40, 10% glycerol in DNase/RNase-free water) with protease inhibitors (Aprotinin, Pepstatin A, Leupeptin, 1 mM PMSF) for 5 min on ice. Nuclei were pelleted by spinning cells at 845xg for 5 min at 8°C, then lysed with 500 µl SDS lysis buffer (50 mM Tris-HCl + 10 mM EDTA + 1% SDS in sterile water) for 5 min on ice. The chromatin from lysed nuclei in SDS lysis buffer was sheared to obtain 200–500 bp DNA fragments using a Diagenode Bioruptor with 35 cycles (30 s on/off) submerged in cold water. Fragment sizes were determined by running a 20 µl aliquot of reverse-crosslinked chromatin on a 1.5% agarose gel. The sheared chromatin was diluted 1:10 with ChIP dilution buffer (50 mM Tris-HCl, 2 mM EDTA, 0.5M NaCl, 0.1% SDS in sterile water) then pre-cleared with washed Dynabeads Protein G for 1 hr at 4°C. The Dynabeads were magnetically isolated from chromatin and 1% of chromatin was separated and reverse crosslinked to be used as input. The remaining sample was incubated with Nkx2-1 antibody (Millipore 07601) overnight at 4°C. To

isolate antibody-bound DNA, washed Dynabeads Protein G were added to each sample (50 ul beads/sample) and incubated for 30 min at 4℃.The Dynabeads with antibody-bound chromatin were isolated magnetically and subjected to 3x washes each with wash buffer (10 mM Tris-HCl, 2 mM EDTA, 0.5M NaCl, 0.1% SDS, 1% NP-40 in sterile water), LiCl buffer (10 mM Tris-HCl, 2 mM EDTA, 0.5M LiCl, 0.1% SDS, 1% NP-40), and TE buffer (1 mM Tris-HCl, 1 mM EDTA) for 5 min on ice. The chromatin was eluted in 100 μl of 2% SDS in TE on a 65℃ heatblock with vigorous shaking (1400 rpm) for 15 min. Input DNA and immunoprecipitated DNA were reverse crosslinked by adding 5 μl 5M NaCl to 100 μl eluate and incubating overnight at 65℃, followed by a 30 min treatment with RNase. The reverse crosslinked DNA was purified using MicroChIP Diapure columns (Diagenode, cat# C03040001) and eluted in 10 μl of elution buffer. The entire eluate of Nkx2-1 immunoprecipitated DNA and 0.5 ng of input DNA were used to prepare ChIP libraries. The libraries were prepared using the Microplex Library Preparation Kit v2 (Diagenode, cat# C05010012) according to manufacturer's instructions with 14 amplification cycles. Library quality and size were calculated using an Agilent 2100 Bioanalyzer with the High Sensitivity DNA kit (Agilent, cat# 5067–4626), and library concentration was quantified with the QuBit dsDNA HS Assay kit (Invitrogen, cat# Q32854). The libraries were pooled to 5 nM and sequenced in one lane of an Illumina HiSeq 4000. The sequencing data were processed and downstream analysis was performed as described below.

## Analysis of scRNA-seq data

We used the Cell Ranger v2.1.1 pipeline from 10X Genomics for initial processing of raw sequencing reads. Briefly, raw sequencing reads were demultiplexed, aligned to the mouse genome (mm10), filtered for quality using default parameters, and UMI counts were calculated for each gene per cell. Filtered gene-barcode matrices were then analyzed using the Seurat v3.0 R package (*Stuart et al., 2018*). Seurat objects were generated with CreateSuratObject (min.cells = 10, min.features = 200) for E10.5 foregut and lung cells, E11.5 foregut cells, and E11.5 lung cells. E11.5 foregut and lung cells were merged to create a single gene-barcode matrix. Cells were further filtered based on the distribution of number of genes (nFeature) and percent mitochondrial genes (percent.mito) per cell across the dataset as follows. nFeature_RNA (E10.5):>2000,<7000, nFeature_RNA (E11.5):>2500, <8500, percent.mito:>0.5,<7.5. Data were normalized for sequencing depth, log-transformed, and multiplied by a scale factor of 10000 using the default parameters of NormalizeData. Linear regression was performed to eliminate variability across cell cycle stage (CellCycleScoring) and mitochondrial content using ScaleData. For E11.5 merged foregut and lung, nCount_RNA was also regressed out as these datasets retained slight variability in sequence depth that was not eliminated with Scale-Data. The top 2000 variable genes within each dataset were selected based on a variance stabilizing transformation (FindVariableGenes, selection.method = 'vst') and used in downstream principal component analysis (PCA). The principal components (PCs) were identified with RunPCA and PCs to include in downstream analysis were empirically determined with visualization of PCs in an Elbow-Plot. Cell clusters were identified by construction of a shared nearest neighbor graph (FindNeighbors) and a modularity optimization-based clustering algorithm (FindClusters) using the PCs determined by PCA (E10.5 dims = 1:20, E11.5 dims = 1:12). Clustering was performed at multiple resolutions between 0.2 and 2, and optimal resolution was determined empirically based on the expression of known population markers and the FindMarkers function (E10.5 resolution = 0.55, E11.5 resolution = 0.45). Several outlying clusters of mesenchymal contamination were removed, and cells were re-clustered for visualization purposes. Cells and clustering were visualized using Uniform Manifold Approximation and Projection (UMAP) dimensional reduction (RunUMAP). Markers for each cluster were identified with FindAllMarkers using default parameters, and cluster identity was determined based on the presence of known markers, as well as experimental evidence of RNA localization in specific cell types.

## Analysis of bulk RNA-seq data

Analysis of RNA-seq reads was performed as described previously (*Percharde et al., 2018*). Differential expression analysis (*Nkx2-1*[-/-] vs WT, WT trachea vs WT esophagus) was performed using DESeq2 (*Love et al., 2014*) (test = c('Wald'), betaPrior = T) and genes with a log2 fold change > 0.7 or<−0.7 and an adjusted p-value<0.05 were determined to be differentially expressed. Differential expression results were visualized using the ggplot2 package.

## Analysis of ChIP-seq data

FASTQ files of raw sequencing reads for NKX2-1 ChIP and input libraries were processed using a custom script (github.com/akelakuwahara/foregut/run ChIPseq). Quality and length trimming and generation of fastqc files to examine sequence quality were performed using Trim Galore (*Krueger, 2014*). Trimmed reads were aligned to the mouse genome (mm9) using bowtie2 and sorted deduplicated bam files were generated using samtools. Peak calling was performed with MACS2 (*Zhang et al., 2008*) using a false discovery rate less than 1e-5 (macs2 callpeak -t chip. sorted.bam -c input.sorted.bam -f BAM -q 0.00001 g mm -n nkx_peaks –outdir macs/). Peaks shared between both replicates were identified by finding the intersection of both replicates using the Intersect tool in Galaxy (usegalaxy.org). Motif analysis to test for the enrichment of the NKX2-1 motif was performed with MEME ChIP using a 500 bp region flanking the peak summit for all peaks shared between both replicates. NKX2-1 binding at specific loci was visualized in the Integrative Genomics Viewer. Peak-gene associations were generated with GREAT using the basal-plus-extension rule (*McLean et al., 2010*).

## Code availability

All code used for data analysis is available at https://github.com/akelakuwahara/foregut (*Kuwahara, 2020*; copy archived at https://github.com/elifesciences-publications/foregut).

## Acknowledgements

We thank Marta Losa for technical advice and guidance on ChIP-seq protocols. We thank Licia Selleri, Ophir Klein, Nadav Ahituv and our laboratory colleagues for constructive feedback on the project and comments on the manuscript. FACS was carried out at the UCSF Parnassus Flow Cytometry Core. Sequencing of bulk RNA and ChIP libraries was carried out by the UCSF Center for Advanced Technology. Single-cell capture and RNA sequencing was carried out by the UCSF Institute for Human Genetics. Computational analysis was performed on the XSEDE computing resource under NSF GRFP grant #1650113.

## Additional information

### Funding

| Funder | Grant reference number | Author |
|---|---|---|
| National Heart, Lung, and Blood Institute | R01HL144785 | Jeffrey Ohmann Bush |
| National Science Foundation | GRFP1650113 | Akela Kuwahara |
| California Institute for Regenerative Medicine | EDUC2-08391 | Coohleen Coombes |

The funders had no role in study design, data collection and interpretation, or the decision to submit the work for publication.

### Author contributions

Akela Kuwahara, Conceptualization, Data curation, Formal analysis, Funding acquisition, Validation, Investigation, Writing - original draft, Writing - review and editing; Ace E Lewis, Formal analysis, Validation, Investigation; Coohleen Coombes, Formal analysis, Investigation; Fang-Shiuan Leung, Investigation, Visualization; Michelle Percharde, Software, Formal analysis, Methodology; Jeffrey O Bush, Conceptualization, Data curation, Formal analysis, Supervision, Funding acquisition, Investigation, Writing - original draft, Project administration, Writing - review and editing

### Author ORCIDs

Jeffrey O Bush https://orcid.org/0000-0002-6053-8756

## Ethics

Animal experimentation: All animal procedures were performed at the University of California San Francisco (UCSF) under approval from the UCSF Institutional Animal Care and Use Committee (protocol # AN164190).

## Decision letter and Author response

Decision letter https://doi.org/10.7554/eLife.55526.sa1
Author response https://doi.org/10.7554/eLife.55526.sa2

# Additional files

## Supplementary files

• Transparent reporting form

## Data availability

Source data for Figure 1, 2, 3, S1-4 are included as supplemental data files to the manuscript. All sequencing data has been uploaded to the Dryad (https://doi.org/10.7272/Q6WW7FVB).

The following dataset was generated:

| Author(s) | Year | Dataset title | Dataset URL | Database and Identifier |
|---|---|---|---|---|
| Kuwahara A, Lewis A, Coombes C, Leung F-S, Percharde M, Bush JO | 2020 | Delineating the early transcriptional specification of the mammalian trachea and esophagus; Expression matrices for scRNA-seq data | https://doi.org/10.7272/Q6WW7FVB | Dryad Digital Repository, 10.7272/Q6WW7FVB |

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
