## [Decision Letter]

**Acceptance summary:**

This manuscript provides a significant amount of new information about how NKX2.1 and SOX2 act in concert to regulate the esophageal and tracheal developmental programs. A particular strength of the manuscript is the coupling of functional genetic experiments with the sequencing data.

**Decision letter after peer review:**

Thank you for submitting your article "Delineating the early transcriptional specification of the mammalian trachea and esophagus" for consideration by *eLife*. Your article has been reviewed by two peer reviewers, and the evaluation has been overseen by a Reviewing Editor and Edward Morrisey as the Senior Editor. The reviewers have opted to remain anonymous.

The reviewers have discussed the reviews with one another and the Reviewing Editor has drafted this decision to help you prepare a revised submission.

We would like to draw your attention to changes in our revision policy that we have made in response to COVID-19 (https://elifesciences.org/articles/57162). Specifically, we are asking editors to accept without delay manuscripts, like yours, that they judge can stand as *eLife* papers without additional data, even if they feel that they would make the manuscript stronger. Thus, the revisions requested below only address clarity and presentation.

Summary:

This manuscript provides new information about how NKX2.1 and SOX2 act in concert to regulate the esophageal and tracheal developmental programs. Overall the manuscript is compelling in that the conclusions are well supported by strong single cell sequencing and mouse genetic experiments.

Revisions:

1) Provide information on numbers for mouse genetic experiments. How many replicate embryos were examined, and how much heterogeneity in phenotype there is across embryos. It will be important to report how many embryos were examined in each case. It will also be helpful if some quantification of the phenotypes is presented – i.e. what proportion of the embryos examined displayed a similar phenotype?

2) Scale bars are generally not reported or lack units – both of these will be helpful for the reader.

i.e. IF/FISH images lack scale bars. ChIP-seq data lack scale and are important for judging the size of peaks, especially between different genes, heatmaps (i.e. Figure 1C) have a scale bar, but units are not reported – are these z-scores?

3) Provide code for the analysis.

---

## [Author Response]

Revisions:1) Provide information on numbers for mouse genetic experiments. How many replicate embryos were examined, and how much heterogeneity in phenotype there is across embryos. It will be important to report how many embryos were examined in each case. It will also be helpful if some quantification of the phenotypes is presented – i.e. what proportion of the embryos examined displayed a similar phenotype?

We have updated figure legends to reflect replicate numbers for each experiment and have added descriptions in the text of mutant phenotypes, which were fully penetrant in every genetic model.

2) Scale bars are generally not reported or lack units – both of these will be helpful for the reader.i.e. IF/FISH images lack scale bars. ChIP-seq data lack scale and are important for judging the size of peaks, especially between different genes, heatmaps (i.e. Figure 1C) have a scale bar, but units are not reported – are these z-scores?

IF/FISH images: we have added units to the scale bar in each image panel, and have indicated in the figure legend that all images within an image panel are taken at the same magnification and displayed at the same scale.

ChIP data: We have included a scale bar indicating the horizontal scale (in kb) of each displayed track, as well as the IGV data range on the y-axis. We have also indicated within the figure legend that the data range for the input and NKX2.1 IP tracks displayed for each gene are scaled equivalently.

Heatmaps: We have included units (z-score) on all heatmaps (Figure 1C, Figure 1—figure supplement 2).

3) Provide code for the analysis.

We have included a section of “Code availability” within the Materials and methods portion of the paper that includes a link to the Github page with the code files for analyses performed in this manuscript.